# Scoping review protocol: The chrononutrition factors in association with glycemic outcomes in adult population

Guey Yong Chong[1], Satvinder Kaur[2], Ruzita Abd Talib[3], See Ling Loy[4,5], Hui Yin Tan[1], Kok Hoe Wilfred Mok[4,6], Ling-Wei Chen[7,8], Woan Yie Siah[9], Yin Yin Chee[1], Ee Mun June Lem[1], Hui Chin Koo[1]*

1 Faculty of Applied Sciences, Tunku Abdul Rahman University of Management and Technology, Kuala Lumpur, Malaysia, 2 Faculty of Applied Sciences, UCSI University, Kuala Lumpur, Malaysia, 3 Faculty of Health Sciences, Nutritional Sciences Program, Centre for Community Health Studies (ReaCH), Universiti Kebangsaan Malaysia, Kuala Lumpur, Malaysia, 4 Department of Reproductive Medicine, KK Women's and Children's Hospital, Singapore, Singapore, 5 Duke-NUS Medical School, Singapore, Singapore, 6 Institute for Health System Research, National Institutes of Health, Centre for Health Services Research, Ministry of Health Malaysia, Putrajaya, Malaysia, 7 Institute of Epidemiology and Preventive Medicine, College of Public Health, National Taiwan University, Taipei, Taiwan, 8 Master of Public Health Program, College of Public Health, National Taiwan University, Taipei, Taiwan, 9 Klinik Kesihatan Batu Berendam, Pejabat Kesihatan Daerah Melaka Tengah, Melaka, Malaysia

* koohc@tarc.edu.my

**Data Availability Statement:** No datasets were generated or analysed during the current study. All relevant data from this study will be made available upon study completion.

## Abstract

Chrononutrition, which examines the relationship between circadian rhythms and nutrition, has been associated with glycemic outcomes in adults. However, published data on delayed meal timing, increased meal frequency and frequent breakfast skipping have shown inconsistent glycemic outcomes due to variations in methodologies and populations studied. This review presents the scoping review protocol designed to map the evidence on the association between chrononutrition factors and glycemic outcomes in adults. The methodology framework from Arksey and O'Malley will be adapted for this scoping review. Relevant publications will be searched on databases including PubMed, EBSCO Host, ProQuest Central, MEDLINE & Ovid, Scopus and Web of Science. This review focuses on original articles published from January 2014 to 2024, involving participants aged 18 years and older, published in English, and encompassing experimental and observational studies. A comprehensive keyword search strategy will be developed to identify relevant articles. Two reviewers will independently screen the abstracts and titles to determine the eligibility. Subsequently, the full text of potentially eligible articles will be reviewed by additional independent reviewer for final inclusion, with full text screening being verified by two reviewers, and interrater reliability will be conducted. Data from the included articles will be extracted, collated and charted to summarize the relevant methods, outcomes and key findings. This Preferred Reporting Items for Systematic Reviews and Meta-Analyses extension for Scoping Reviews (PRISMA-ScR) checklist will be used to guide the development of protocol. This scoping review represents a novel approach to summarize the association between chrononutrition factors and glycemic outcomes among adults. We anticipate the findings of the review will provide stakeholder with crucial evidence-based information for development of effective

**Funding:** The research is funded by the Ministry of Higher Education Malaysia, Fundamental Research Grant Schema (FRGS/1/2021/SKK06/TARUC/02/1). The funders had no role in the study design; collection, management, analysis and interpretation of data; preparation of the manuscript decision, or submission of the report for publication.

**Competing interests:** None. The authors have declared that there are no competing interests.

**Abbreviations:** HbA1c, Glycated hemoglobin; IRR, Interrater reliability; KNHANES, Korea National Health and Nutrition Examination; MeSH, Medical Subject Heading; MMAT, Mixed Methods Appraisal Tool; NHANES, National Health and Nutrition Examination Survey; PEO, Popolation, Exposure, Outcomes; PRISMA-ScR, Preferred Reporting Items for Systematic Reviews and Meta-Analyses Extension for Scoping Reviews; SPSS, Statistical Package for the Social Sciences; US, United State.

intervention to manage glycemic outcome in adults. This protocol has been prospectively registered in the Open Science Framework (https://doi.org/10.17605/OSF.IO/PA9BU).

## Introduction

Diabetes is a lifelong metabolic disease with a rising global prevalence, projected to affect 260.2 million adults in the Western Pacific region by 2045 [1,2]. Despite the American Diabetes Association's lifestyle recommendations have suggested healthy diet and physical activity to regulate blood glucose levels, but current guidelines may be insufficient for effective diabetes management [3]. Emerging evidence increasingly highlights the critical role of circadian clock in metabolic regulation, introducing chrononutrition as a significant factor in diabetes management [4–6]. Recently, modern adults often engage in irregular eating patterns, including skipping breakfast [7,8], extending daily eating windows [9], reducing nocturnal fasting periods [10], and frequent nighttime snacking [11]. These behaviors are closely related to insulin resistance, leading to elevated glycated hemoglobin levels (HbA1c), increased postprandial and fasting plasma glucose levels, and a higher risk of diabetes [12–15].

Chrononutrition's key components, including energy distribution, meal regularity and frequency of meals, have been shown to influence glycemic outcomes [16]. Current evidence had extended its components to meal timing, specifically night eating [17]. Large-scale studies have demonstrated the association between eating later in the day and poor glycemic control [14,18]. For example, the National Health and Nutrition Examination Survey (NHANES) in the United State (US) found a significant association between the last meal time before bed and elevated HbA1c levels[14]. However, findings from the Korea National Health and Nutrition Examination (KNHANES) did not show a significant association between evening eating, defined as consuming more than 40% of daily energy between 6.00pm to 9.00pm, and adverse glycemic outcomes [18]. In contrast, a US study suggested that restricting eating to window from 4.00 am to 4.00 pm, with light dinner and energy distribution focused on breakfast and lunch could stabilize glucose levels and reduce fluctuations [19]. This statement is also in agreement with Asian studies, which indicated that hyperglycemia is associated with eating meals after 8.00pm [20,21]. Studies also indicated that fasting gaps allocated before sleep by refraining from late-night dinner eating may regulate blood glucose effectively [22,23]. However, identifying specific time-related dietary factors that contribute to glucose intolerance remains complex, influenced by seasonal variations, daylight exposure and individual lifestyle factors [24–26].

Given meal timing is related to meal frequency, night fasting duration, and energy distribution of a day, this study explores their associations with glycemic outcomes. Previous studies have reported that irregular meals, specifically eating more than four meals or fewer than two meals daily, are associated with an increased risk of diabetes [27–30]. However, other studies have found no significant association between meal frequency and diabetes risk [10,31]. These discrepancies are likely due to variations in meal timing and frequencies, leading to inconsistent glycemic outcomes [28,32]. A systematic review suggests that restricting eating to 2 to 3 meals daily, within a feeding window of less than 10 hours and consuming the last meal 3 to 4 hours before sleep, may effectively manage glucose levels in individuals with type 2 diabetes [33]. However, this strategy's applicability to other populations, including healthy individuals [34], those who are overweight or obese [35], where the effects of meal timing remain uncertain. Furthermore, breakfast skippers delayed their first meal and reserved majority of their

energy intake at evening time has provided mixed results on glycemic outcomes [36,37]. Thus, the available evidence showed inconsistency in the association between chrononutrition factors and glycemic outcomes.

Conventional methods for diagnosing glycemic outcomes, such as random blood glucose, fasting plasma glucose, the 75g oral glucose tolerance test and HbA1c levels may not reliably capture advanced interday or intraday glucose fluctuations [38]. Therefore, this study incorporates studies that using the ambulatory glucose profiles derived from continuous glucose monitoring sensors could provide a more advanced and comprehensive understanding of 24-hour glucose variability in individuals with hyperglycemia [39]. Understanding how chrononutrition factors interact and influence glycemic outcomes is essential, although health effects of combining different chrononutrition factors remain uncertain. A comprehensive evaluation of these factors could provide valuable insights into their association with glycemic outcomes. With the ultimate aim to improve the diabetes management, the present review will identify and analyzes the knowledge gaps regarding the association between chrononutrition factors and glycemic outcomes in adult population. Specifically, this review aims to:

1. investigate the association between chrononutrition factors (meal regularity, meal timing and meal frequency) and glycemic outcomes in terms of glucose indices, insulin indices and incidence of diabetes among adults.

2. describe the chrononutrition factors (meal regularity, meal timing and meal frequency) applied in various populations and their effect on glycemic outcomes (glucose indices, insulin indices and incidence of diabetes) of the populations.

3. examine the opportunities to integrate chrononutrition into dietary guideline to prevent and control diabetes mellitus.

## Materials and methods

The scoping review will follow the methodological framework proposed by Arksey and O'Malley [40], extended by Levac et al. [41] and with further refinement by the Joanna Briggs Institute [42]. Specifically, five steps of methodology will be adopted in our scoping review in Fig 1.

Due to time and budget constraints, the consultation with relevant experts or stakeholders to generate insights will not be included in the present review. Verhage and Boels (2016) acknowledge the role of quality appraisal in scoping review [43]. In this context, the Mixed Methods Appraisal Tool (MMAT 2018) will be adopted to assess the quality of the included studies [44], while the PRISMA-ScR checklist will be adopted in Fig 2 [45]. The protocol is registered with the Open Science Framework (OSF registration: https://doi.org/10.17605/OSF.IO/PA9BU). Any modifications to the protocol will be documented in OSF. Prior to protocol registration, the first reviewer has found no similar title appear in OSF and Cochrane.

### Step 1: Identification of research questions

The first step of the protocol is to establish a research question by seven reviewers. The research question seeks to clarify and describe the linkage of the study purpose and ideas comprehensively. The research question will use the PEO framework in Table 1 to construct a research question [46]. "P" is the population; "E" is the exposure, and "O" is the outcome. In this review, our "P" refers to adults aged 18 to 60 years. We considered individuals aged 18 to 60 years in this group represents a relatively stable period of physical and physiological functioning, minimizing the influence of aging-related factors that could complicate data

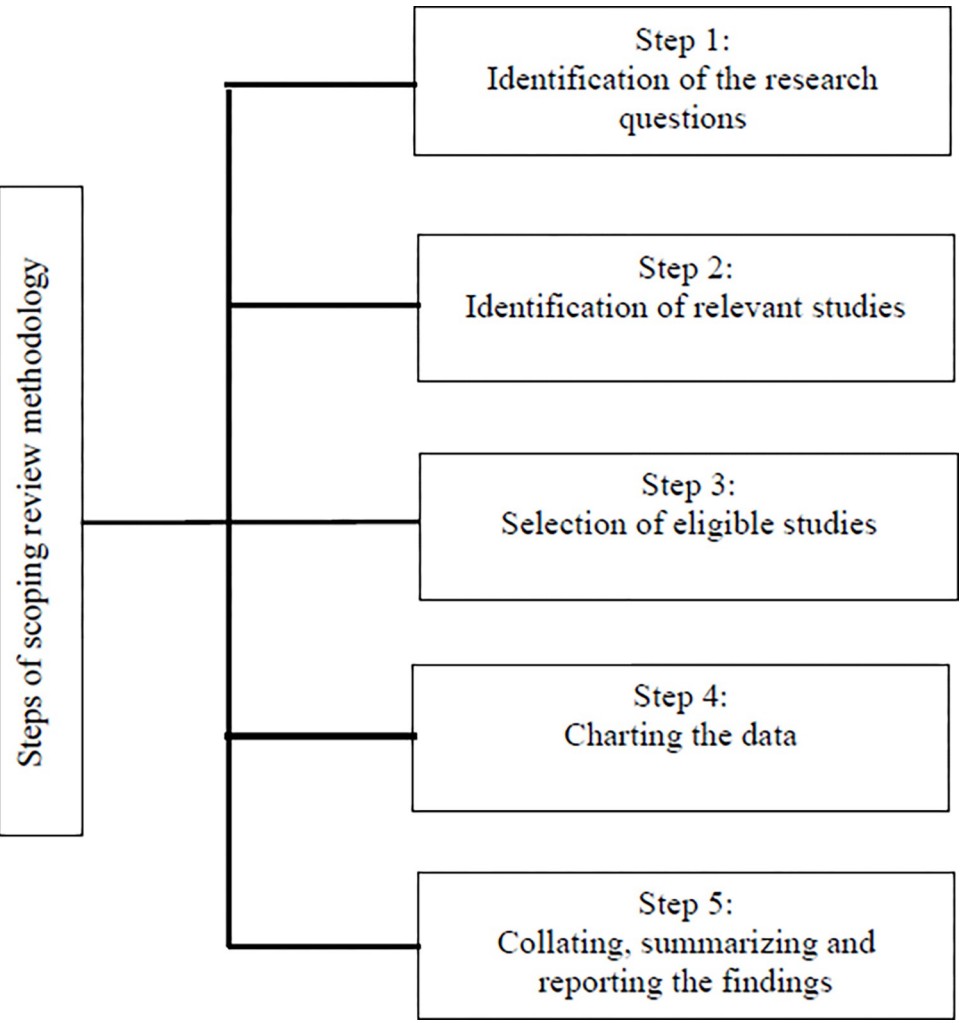

**Fig 1. Steps of scoping review methodology.**

interpretation. This age group also aligns with the working-age population, which typically has consistent lifestyle patterns related to diet, physical activity and sleep [47,48]. Exposure "E" is the chrononutrition factors, such as meal frequency, meal skipping, meal timing, distribution of energy and macronutrients daily, durations of eating and night fasting. The primary outcome "O" is glucose outcomes (glycated hemoglobin, fasting plasma glucose, postprandial glucose and ambulatory glucose profile from continues glucose monitoring system). The secondary outcome is insulin outcomes (insulin sensitivity, insulin resistance, c-peptide function and insulin secretion). The tertiary outcome is the incidence of diabetes. The main research questions are developed according to the scientific gaps, defined as:

1. Are chrononutrition factors associated with the elevation of glycemic outcomes among adults?

2. Is there any positive effect of chrononutrition factors on glycemic outcomes among adults?

3. Are chrononutrition factors possible to integrate into dietary guideline in diabetes management?

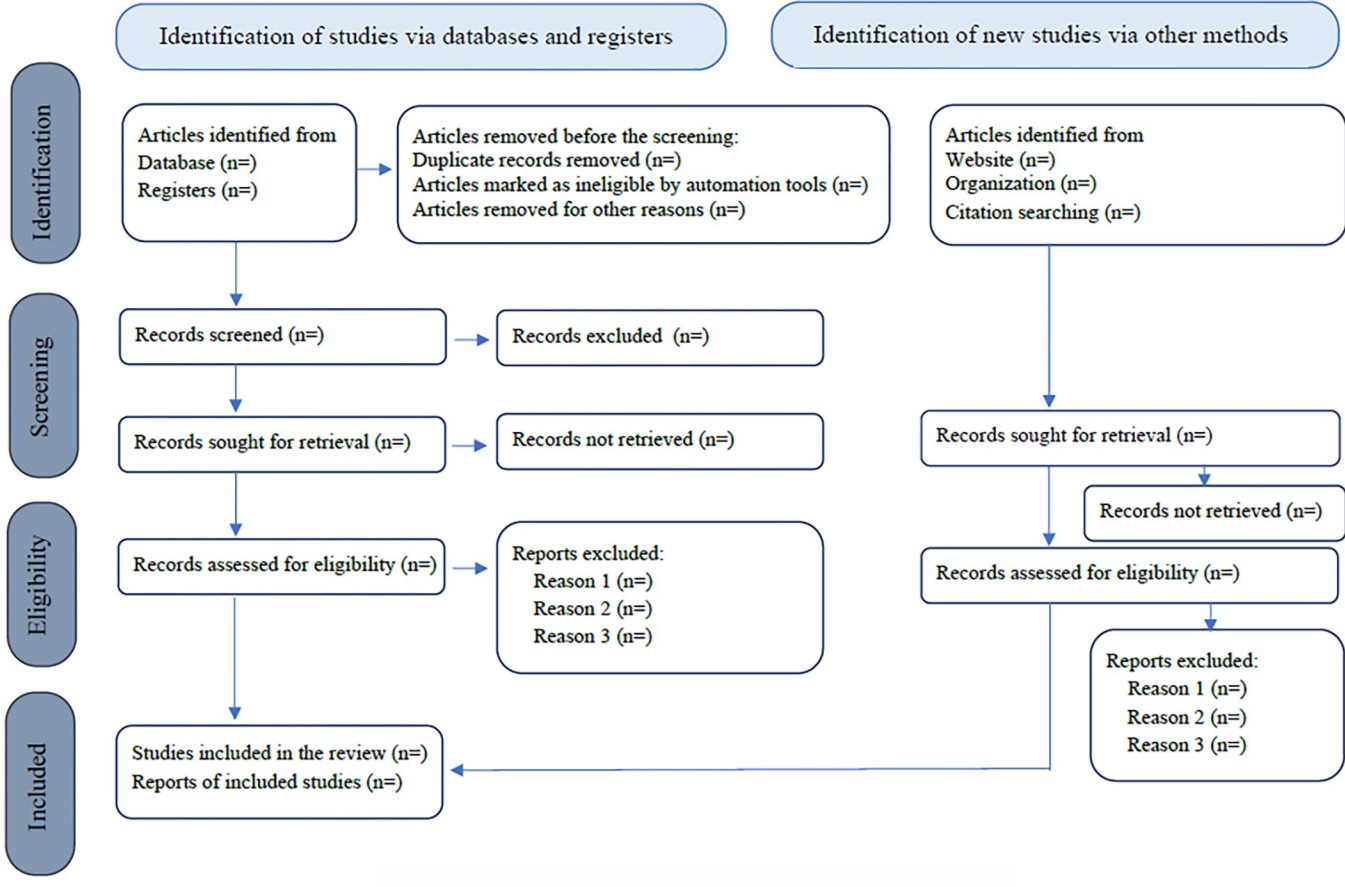

**Fig 2. Flow chart adopted from PRISMA-ScR.**

## Step 2: Identification of relevance studies

Identification of the relevance studies will be achieved by searching the following electronic databases: (1) PubMed, (2) EBSCO Host, (3) ProQuest Central, (4) MEDLINE & Ovid and (5) Web of Science. An initial exploratory search based on the PEO framework will be developed to capture relevant terms and studies by the first reviewer. Boolean terms, AND/OR will be applied to distinguish the keywords. The Medical Subject Heading (MeSH) will be found in the search. The other databases will combine search terms from two themes (1)

**Table 1. PEO framework for determining eligibility of the research question.**

| P-Population | Adults aged 18 to 60 years |
|---|---|
| E-Exposure | 1) Meal frequency<br>2) Meal skipping<br>3) Meal timing<br>4) Night fasting duration<br>5) Distribution of energy and macronutrients/day |
| O-Outcome | 1) Glucose outcomes (glycated hemoglobin, fasting plasma glucose, postprandial glucose and ambulatory glucose profile from continues glucose monitoring system)<br>2) Insulin outcomes (insulin sensitivity, insulin resistance, c-peptide function and insulin secretion)<br>3) Incidence of diabetes |

chrononutrition factors e.g. meal timing, meal frequency, meal skipping, eating window, eating occasion, fasting period, distribution of energy and macronutrients daily; (2) glycemic outcomes, insulin outcomes and incidence of diabetes, to retrieve relevant studies. A reviewer will visually scan a reference list of relevant studies to identify the additional relevant studies, with assistance of two reviewers.

The comprehensive keyword search strategy will be guided by an experienced reviewer and finalized by the first reviewer. The inclusion and exclusion criteria will be refined iteratively, to align potential studies that fulfil the objective of the present scoping review. Studies that focus on: (1) adult population aged 18 to 60 years; (2) original journal articles published from 2014 to 2024; (3) the full text available in English will be included and (4) the experimental and observational studies. This review will exclude: (1) those pregnant and lactating women, shift workers, children and adolescents; (2) type I diabetes mellitus patients; (3) non-human subjects or in vitro studies and (4) abstracts, conference proceedings, editorial articles, case reports, short or brief reports, pilot studies, thesis, reviews, unpublished research or any articles not freely available in full text.

The pilot search strategy on PubMed and other electronic databases will be carried out for relevant studies searching, to identify the appropriate keywords in title, abstract and index terms. The keywords from two themes will be searched in several electronic databases and separated by Boolean operators AND and OR. The selected electronic databases included PubMed, EBSCO Host, ProQuest Central, Medline & Ovid and Web of Science. Hand-searching will be performed to supplement structured searches in the databases [49]. The complete search strategy and the result of pilot searches in the several electronic databases are presented in Table 2.

## Step 3: Selection of eligible studies

In the studies selection step, three reviewers will independently perform the two-stage screening: (1) initial title and abstract screening and (2) full-text review. The search results will be downloaded into a Microsoft Excel file and duplicates will be removed. Given the initial searches yielded 491 records, the title and abstract will be screened according to the eligibility criteria outlined above. The first reviewer will lead the abstract screening process, with another reviewer involve in defining the review aims, scope and screening criteria. Both reviewers independently categorize each study into the category of 'include' or 'exclude'. The included title and abstract will be discussed if there are discrepancies in study selection until a consensus is reached. If the pair reviewers do not reach a consensus, the screening team will discuss and make a final decision.

The second stage of full-text review will start after the initial screening is completed. In the second stage, the first reviewer and additional reviewer will investigate the full-texts of the potentially eligible studies, independently to verify that the eligibility criteria are met. Subsequently, interrater reliability (IRR) will be established to assess the degree of agreement of two reviewers [50]. Cohen's kappa coefficient ($\kappa$) statistic will be performed using the Statistical Package for the Social Sciences (SPSS) version 22 to derive the inter-rater agreement between the two reviewers at the end of the review stage. The value of Cohen's kappa coefficient of at least 0.80 and above represents a strong level of agreement [51]. If the Cohen's kappa coefficient below 0.8, 10% of the randomly selected studies will be screened by same two reviewers and the process will be repeated. The reasons for the exclusion of the studies will be documented for full-texts review. A PRISMA-ScR flow chart will be adopted to report the screening results, as illustrated in Fig 2 [45].

**Table 2. The complete searching strategy and pilot searches in several electronic databases.**

| Database | Themes | Keywords | Searching Strategy | Results |
|---|---|---|---|---|
| PubMed | Chrononutrition | A1. "chrononutrition"[tw]<br>A2. "chrono-nutrition dietary pattern*"[tw]<br>A3. "circadian eating"[tw] | A29. A1 OR A2 OR A3 OR A4 OR A5 OR A6 OR A7 OR A8 OR A9 OR A10 OR A11 OR A12 OR A13 OR A14 OR A15 OR A16 OR A17 OR A18 OR A19 OR A20 OR A21 OR A22 OR A23 OR A24 OR A25 OR A26 OR A27 OR A28 | 772 |
| | Meal regularity | A4. "meal regularity"[tw]<br>A5. "irregular intake"[tw] | | |
| | Meal timing | A6. "night eating syndrome"[Mesh]<br>A7. "late-night-dinner"[tw]<br>A8. "meal timing"[tw]<br>A9. "late-night eating"[tw]<br>A10. "night-time snacking"[tw]<br>A11. "night eating"[tw]<br>A12. "food timing"[tw]<br>A13. "late eating"[tw]<br>A14. "timing of food"[tw]<br>A15. "evening snacking"[tw] | | |
| | Meal frequency | A16. "meal frequency"[tw]<br>A17. "eating frequency"[tw] | | |
| | Meal skipping | A18. "meal skipping"[tw]<br>A19. "breakfast skipping"[tw] | | |
| | Eating window | A20. "eating window"[tw]<br>A21. "time-restricted eating"[tw]<br>A22. "eating episode"[tw] | | |
| | Eating occasion | A23. "eating occasion"[tw]<br>A24. "meal occasion"[tw] | | |
| | Fasting period | A25. "Fasting interval"[tw]<br>A26. "Intermediate fasting"[tw] | | |
| | Distribution of energy and macronutrients | A27. "energy distribution"[tw]<br>A28."macronutrient* distribution"[tw] | | |
| | Glycemic Outcomes | B1. "prediabetes"[tw]<br>B2. "metabolic syndrome"[tw]<br>B3. "Mets"[tw]<br>B4. "Type 2 diabetes"[tw]<br>B5. "glucose outcome*"[tw]<br>B6. "diabetes management"[tw] | B21. B1 OR B2 OR B3 OR B4 OR B5 OR B6 OR B7 OR B8 OR B9 OR B10 OR B11 OR B12 OR B13 OR B14 OR B15 OR B16 OR B17 OR B18 OR B19 OR B20 | 49912 |
| | Glycated hemoglobin | B7. "Glycated Hemoglobin"[Mesh]<br>B8. "hba1c"[tw] | | |
| | Fasting plasma glucose | B9. "Blood glucose"[Mesh]<br>B10. "fasting plasma glucose"[tw]<br>B11. "hyperglycemia"[tw] | | |
| | 2-hour postprandial glucose | B12. "postprandial"[tw]<br>B13. "nocturnal glucose"[tw] | | |
| | Glucose variability | B14. "glucose variability"[tw]<br>B15. "glycemic variability"[tw] | | |
| | Insulin | B16. "Insulin"[Mesh]<br>B17. "insulin resistance"[tw]<br>B18. "insulin sensitivity"[tw]<br>B19. "insulin secretion"[tw]<br>B20. "c-peptide"[tw] | | |
| | | | A29 AND B21 | 187 |

(*Continued*)

**Table 2.** (Continued)

| Database | Themes | Keywords | Searching Strategy | Results |
|---|---|---|---|---|
| EBSCO Host, ProQuest Central, Medline & Ovid, Web of Science | Chrononutrition | A1. chrononutrition<br>A2. chrono-nutrition dietary pattern*<br>A3. circadian eating | A29. A1 OR A2 OR A3 OR A4 OR A5 OR A6 OR A7 OR A8 OR A9 OR A10 OR A11 OR A12 OR A13 OR A14 OR A15 OR A16 OR A17 OR A18 OR A19 OR A20 OR A21 OR A22 OR A23 OR A24 OR A25 OR A26 OR A27 OR A28 | 10370 |
| | Meal regularity | A4. meal regularity<br>A5. irregular intake | | |
| | Meal timing | A6. night eating syndrome<br>A7. late-night-dinner<br>A8. meal timing<br>A9. late-night eating<br>A10. night-time snacking<br>A11. evening snacking<br>A12. night eating<br>A13. food timing<br>A14. late eating<br>A15. timing of food | | |
| | Meal frequency | A16. meal frequency<br>A17. eating frequency | | |
| | Meal skipping | A18. meal skipping<br>A19. breakfast skipping | | |
| | Eating window | A20. eating window<br>A21. time-restricted eating<br>A22. eating episode | | |
| | Eating occasion | A23. eating occasion<br>A24. meal occasion | | |
| | Distribution of energy and macronutrients | A25. energy distribution A26. macronutrient* distribution | | |
| | Fasting period | A27. Fasting interval<br>A28. Intermediate fasting | | |
| | Glycemic Outcomes | B1. prediabetes<br>B2. metabolic syndrome<br>B3. Mets<br>B4. Type 2 diabetes<br>B5. glucose outcome*<br>B6. diabetes management | B21. B1 OR B2 OR B3 OR B4 OR B5 OR B6 OR B7 OR B8 OR B9 OR B10 OR B11 OR B12 OR B13 OR B14 OR B15 OR B16 OR B17 OR B18 OR B19 OR B20 | 200498 |
| | Glycated hemoglobin | B7. Glycated Hemoglobin A<br>B8. hba1c | | |
| | Fasting plasma glucose | B9. Blood glucose<br>B10. fasting plasma glucose<br>B11. hyperglycemia | | |
| | 2-hour postprandial glucose | B12. postprandial<br>B13. nocturnal glucose | | |
| | Glucose variability | B14. glucose variability<br>B15. glycemic variability | | |
| | Insulin | B16. Insulin<br>B17. Insulin resistance<br>B18. Insulin secretion<br>B19. Insulin sensitivity<br>B20. C-peptide | | |
| | | | A29 AND B21 | 285 |
| Hand searching | | | A29 AND B21 | 19 |
| Search conducted on 10 August 2024 | | | | 491 |

**Table 3. Overview of the data charting form and coding table.**

| Scoping review charting form | | |
|---|---|---|
| Citation information | Authors | Indicate first author (first name, first name/initial) |
| | Year of publication | Indicate the year of publication |
| | Title of articles | Indicate the full title of the articles |
| | Title of journal | Indicate the title and subtitle of the journal |
| | Website URL | Indicate the website URL if applicable |
| | Abstract | Indicate the abstract of the articles |
| | Keywords | Indicate the keywords of the articles |
| Methodology | Recruitment year | Indicate the start and end of recruitment year |
| | Study location | Indicate all the study locations (e.g. countries, stages, town, hospital, community clinics, etc.) |
| | Sample size | Indicate the sample size calculation, dropout rate and final sample size recruited in study. |
| | Sample characteristics | Indicate gender, race, education background, socioeconomic status, history of diseases, etc. |
| | Age | Indicate the age range, mean of age or target age of population. |
| | Study design | Indicate the study design of the studies |
| | Follow up period or intervention period | Indicate the observation period or the intervention duration |
| Findings and contents | Dietary assessment for chrononutrition variables | Indicate the dietary assessment tools applied in chrononutrition variables |
| | Exposure (definition) | Indicate the definition of chrononutrition factors in the studies. |
| | Outcome (definition) | Indicate the definition of outcomes in the studies. |
| | Confounders or covariates | Indicate the confounders or covariates whether it is controlled or under controlled. |
| | Statistical analysis | Indicate the statistical analysis for primary and secondary outcomes which are related with exposures. |
| | Results | Indicate the results/ findings of the evidence source when applicable. |
| | MMAT (2018) appraisal score | Indicate the appraisal score in each study. |

## Step 4: Charting the data

In Step 4, the results of the eligible studies will be tabulated and recorded in a standardized data chart template, in the form of a Microsoft Office Excel spreadsheet. The data chart form is based on Staden et al. (2022) [52] to collect relevant outcomes and study characteristics of each selected study. The overview of the data charting form is shown in Table 3. The study characteristics will include study metrics, e.g. authors, year of publication, recruitment year, study location, sample size, population characteristics, population's age, study design, follow up period or intervention period, dietary assessments for exposure, definition of exposure, definition of outcomes, covariates or confounders, statistical analysis, results and the appraisal score. The data extraction form will be displayed in Supplementary Material 2. All extracted data from the chart form will be thematically analyzed to facilitate responding to the research question. The first reviewer and one more reviewer will test the data charting form in a pilot with a subset of ten randomly selected studies to ensure that it is useful and appropriate for reporting data. A complete data chart will be summarized by each independent reviewer respectively. The first reviewer will evaluate the data chart for accuracy and completeness, point out any divergence and remark future recommendation. The inconsistencies will be discussed until a consensus is reached. All coding will be completed by two reviewers to minimize systematic or potential bias.

## Step 5: Collating, summarizing and reporting the findings

The findings aim to summarize the association of chrononutrition factors and glycemic outcomes in adult population. In Step 5, the analysis of thematic content will be performed by the

first reviewer and another reviewer after data extraction. The quantitative data will be presented using figures and tables. The data will be qualitatively explained the assessment used, population characteristics, similarities as well as disparate approaches and contextual distinction in relation to the research question. Descriptive statistics will be used to summarize the numerical outcomes across studies, providing an overview of the findings related to association between chrononutrition factors and glycemic outcomes. The authors will explore trends and patterns in the data to identify consistent findings or disparities among the studies. This approach will enable authors to compare and contrast the results, offering a comprehensive understanding of the association between chrononutrition and glycemic outcomes. All findings from the retrieved data will be combined to develop a conclusion in the scoping review. The eligible studies will guide the map of scientific evidence and discover the scientific gaps e.g. authors hypothesize that the late last meal timing, irregular meal intake, long eating duration of a day and short night fasting period are associated with glycemic intolerance and other relevant outcomes.

## Methodological quality appraisal

The purpose of the methodological quality appraisal is to conduct the sensitivity analysis to identify the quality threshold on the original synthesis results using the Mixed-Method Appraisal Tool (MMAT 2018) [44]. The tool is designed to appraise the methodology quality of four classifications of studies (qualitative research, quantitative randomized controlled trials, quantitative non-randomized studies and mixed methods studies). In the present review, the quality appraisal process will use questions from the category of quantitative randomized controlled trials and quantitative non-randomized studies in MMAT 2018. Three reviewers will work independently to identify the risk of bias in selected studies. A reviewer will partner with the first reviewer to evaluate the experimental studies. Additionally, another reviewer will work together with the first reviewer to assess study quality of the observational studies. The reviewers will meet to discuss any disagreements, create consensus and record the final decision. The disagreement will be resolved by the fourth experienced reviewer. Each selected study will be assigned a quality score to assess its quality of methodology. A quality score of $\leq$50%, 51 to 75% and 76 to 100% will be interpreted as low quality, average quality and high quality respectively [53]. The systematic bias and potential errors will be summarized as a narrative statement and supported by a tabulated table.

## Discussion

This scoping review will utilize the transparent and reproducible procedure throughout the review process. The rigorous selection of data sources, search strategy, data extraction, and the quality assessment methods will ensure broad coverage of the existing literature. An assessment of study quality, while not mandatory in scoping reviews, is a strength of this review as it helps identify potential biases in the study findings. Additionally, this quality assessment will serve as a precursor to a full systematic review and meta-analysis. Current evidence on the association between chrononutrition-related factors and glucose outcomes remains inconclusive [16,54–57]. This review aims to elucidate the optimal meal timing, meal frequency, night fasting duration and other chrononutrition factors that can optimize glucose levels and enhance glucose responses.

Addressing the existing scientific gaps, the synthesized evidence will provide valuable guidance for supporting, developing, and implementing effective lifestyle interventions[9]. At the individual level, tailoring food intake timing to align with an individual's biological clock, chronotype and circadian rhythm through personalized chrononutrition interventions could be a feasible and cost-effective approach for regulating abnormal glycemic outcomes [58].

Moreover, integrating chrononutrition interventions with strategies to address food insecurity is essential to ensure consistent meal timing. Combining chrononutrition with other lifestyle interventions, such as those focused on enhancing diet quality or promoting sleep health, could further optimize the health benefits of these approaches [6].

At population level, the findings of the review are crucial for governments and policymakers. Incorporating circadian eating into clinical practice guideline, dietary recommendations and public health policies aimed at managing glycemic control and preventing diabetes could have significant benefits. Specifically, these guidelines might advocate for consistent daily eating start and end times, ensuring the last meal occurs before a certain clock time or within a defined period before bedtime. By identifying the most impactful chrononutrition factors influencing glycemic outcomes, resources can be efficiently directed towards targeted programs and initiatives, thus maximizing the return on investment in public health. Furthermore, this review has also highlighted the gaps in current research, promoting collaboration between government bodies, academic institutions and healthcare providers. This cooperation is essential for driving innovation in chrononutrition and metabolic health research. The insights gained can also facilitate international collaborations, ensuring the country remains at the forefront of advancements in nutrition and public health. By developing of long-term strategies to reduce the burden of metabolic diseases, the review's findings can ultimately enhance the quality of life for the population.

## Limitation

Despite the defined search approach, there are some limitations in this scoping review that are worth noting. First, due to logistical constraint, this review will only include articles written in English. Second, only studies published within the last 10 years will be considered, which may exclude potentially useful sources of evidence. Thirdly, scoping reviews inherently provide broader but less in-depth information on a topic and serve primarily as a hypothesis-generating approach. Although several limitations are stated above, the findings of this scoping review will be attractive for both researchers and policymakers in healthcare management. By systematically outlining existing evidence, the review aims to tackle knowledge gaps related to the association between chrononutrition and glucose tolerance. This review will also provide the groundwork for future systematic reviews and meta-analyses. The findings will be published in a peer-reviewed journal and presented at a scientific conference.

The authors anticipate that the scoping review will elucidate the association between delayed meal timing, frequent meal intake and increased breakfast skipping with glycemic outcomes. By exploring these chrononutrition factors in relation to glycemic control, the review provides valuable insights for the public health sector. This evidence can serve as basis for developing hypotheses that involve the development of evidence-based lifestyle intervention programs and support the integration of chrononutrition principles into national dietary guidelines, ultimately aiding in the prevention and management of metabolic diseases.

## Supporting information

**S1 Checklist. Supplementary materials PRISMA-ScR.**
(DOCX)

## Acknowledgments

The authors acknowledge the helpful comments from peer reviewers. The authors also express their gratitude for the contributions of Suet Kei Wu, Sim Yee Lim, Ying Hui Lee and Qatrun Nabila bt Derani who guided and helped in the early stages of this review.

## Author Contributions

**Conceptualization:** Guey Yong Chong, Satvinder Kaur, Ruzita Abd Talib, See Ling Loy, Kok Hoe Wilfred Mok, Ling-Wei Chen, Hui Chin Koo.

**Funding acquisition:** Hui Chin Koo.

**Methodology:** Guey Yong Chong, Satvinder Kaur, Ruzita Abd Talib, See Ling Loy, Hui Yin Tan, Kok Hoe Wilfred Mok, Ling-Wei Chen, Woan Yie Siah, Yin Yin Chee, Ee Mun June Lem, Hui Chin Koo.

**Supervision:** Satvinder Kaur, Ruzita Abd Talib, See Ling Loy, Hui Yin Tan, Kok Hoe Wilfred Mok, Ling-Wei Chen, Hui Chin Koo.

**Writing – original draft:** Guey Yong Chong.

**Writing – review & editing:** Guey Yong Chong, Satvinder Kaur, Ruzita Abd Talib, See Ling Loy, Hui Yin Tan, Kok Hoe Wilfred Mok, Ling-Wei Chen, Woan Yie Siah, Yin Yin Chee, Ee Mun June Lem, Hui Chin Koo.

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
