## [Decision Letter · Decision Letter 0]

9 Jul 2024

PONE-D-23-29067Scoping review protocol: The chrononutrition factors in association with glycemic outcomes in adult population.

PLOS ONE

Dear Dr. Koo,

Thank you for submitting your manuscript to PLOS ONE. After careful consideration, we feel that it has merit but does not fully meet PLOS ONE’s publication criteria as it currently stands. Therefore, we invite you to submit a revised version of the manuscript that addresses the points raised during the review process.

**The reviewers have suggested further changes are needed before accepting the manuscript.**

We look forward to receiving your revised manuscript.

Kind regards,

Yee Gary Ang, MBBS MPH

Academic Editor

PLOS ONE

 "The authors acknowledge the Ministry of Higher Education of Malaysia for funding the present review. The fund was provided through the Fundamental Research Grant Scheme (FRGS/1/2021/SKK06/TARUC/02/1). Also, thanks go to the helpful comments from peer reviewers. The authors also express their gratitude for the contributions of Suet Kei Wu, Sim Yee Lim, Ying Hui Lee and Qatrun Nabila bt Derani who guided and helped in the early stages of this review. "

"The research is funded by the Ministry of Higher Education Malaysia, Fundamental

Research Grant Schema (FRGS/1/2021/SKK06/TARUC/02/1). The funders had no role

in the study design; collection, management, analysis and interpretation of data;

preparation of the manuscript decision, or submission of the report for publication."

Reviewers' comments:

Reviewer's Responses to Questions

**Comments to the Author**

1. Does the manuscript provide a valid rationale for the proposed study, with clearly identified and justified research questions?

Reviewer #1: Yes

Reviewer #2: Yes

2. Is the protocol technically sound and planned in a manner that will lead to a meaningful outcome and allow testing the stated hypotheses?

Reviewer #1: Yes

Reviewer #2: Partly

3. Is the methodology feasible and described in sufficient detail to allow the work to be replicable?

Reviewer #1: Yes

Reviewer #2: Yes

4. Have the authors described where all data underlying the findings will be made available when the study is complete?

Reviewer #1: No

Reviewer #2: Yes

5. Is the manuscript presented in an intelligible fashion and written in standard English?

Reviewer #1: Yes

Reviewer #2: Yes

6. Review Comments to the Author

You may also provide optional suggestions and comments to authors that they might find helpful in planning their study.

Reviewer #1: The scoping review protocol is an interesting beginning for a research study. I have many minor modifications and few questions about the manuscript. The overall work look interesting and it could provide a good paper to fill the clinical evidence summarizing the information on chrononutrition factors in association with glycemic outcomes in adult population.

The introduction (both in abstract and the manuscript) should be enlarged in order to provide larger detail and context of the topic. Actually is pretty limited.

Why you haven't included grey literature in the studies selected? Please justify it.

Why only studies from 2012? Plese justify it.

In the introduction you report that in 2045 there will be 783 millions of people with diabetes. I could suggest also to introduce the actual incidence and prevalence worldwide or in some specific countries (such as US, Europe).

When you state that "In recent years, a growing body of evidence suggests that the circadian clock interconnects with nutrients, and impacts on body's physiological process, namely chrononutrition [7]", I expect at least several manuscripts and some systematic literature reviews that state this, not just a study.

In the objective of the scoping review you state that you will provide a definition of chrononutrition characteristics. What will you use for it? Maybe a concept analysis? Please provide larger detail

Can you provide in the supplementary material the data extraction form that you will use in the scoping review?

Can you provide larger details of the benefits of this review for government and policymakers?

Why using POE and not PICOTS? Please justify

When will you start the Scoping review?

Please state also in the limitation of the study the fact that you are providing a Scoping review and not a Systematic Literature Review that will be more appropriate. In addition, how will you compare the numerical outcomes? Please state it in the manuscript

Reviewer #2: I am thankful to the authors for giving updated information on The Chrono nutrition factors in association with glycemic outcomes in the adult population. The different parts of this review manuscript is well prepared. This review article can be moved further after fulfilling the necessary suggestions which are listed below

Major comments

1. The discussion is too narrow. The discussion parts fail to meet the standard outcome of this review

2. The level of English must be improved.

3. In the abstract section, the authors cannot justify their argument. It should be more precise about the core findings (should include scientific outcomes)

Minor comments

1. L117, more clearly the objective by adding one or more sentences and also emphasizes using review.

2. L127, 9.00pm to 9.00 pm or 21:00.

3. L147, Materials and methods. The authors are encouraged to add a flowsheet at the end of describing steps (steps 1-5).

4. L214-215, Given the initial searches yielded 627 records…….. In which criteria did authors find those 627 records? Keywords? Which keywords?

5. L306-312, it is better to make another paragraph stating the possible conclusion of this review.

7. PLOS authors have the option to publish the peer review history of their article (what does this mean?). If published, this will include your full peer review and any attached files.

Reviewer #1: No

Reviewer #2: No

---

## [Author Response · Author response to Decision Letter 0]

2 Sep 2024

Dear reviewers from PLOS One, 

We sincerely thank the reviewers for taking the time to review our manuscript and providing constructive feedback to improve our manuscript. We sincerely hope that you may reconsider our manuscript entitled “Scoping review protocol: The chrononutrition factors in association with glycemic outcomes in the adult population.” (Ref No: PONE-D-23-29076) for publication in PLOS One. Enclosed is the revised version of our manuscript. 

Below are our responses to the comments and recommendations.

Reviewers' comments:

Reviewer's Responses to Questions

Comments to the Author

1. Does the manuscript provide a valid rationale for the proposed study, with clearly identified and justified research questions?

Reviewer #1: Yes

Reviewer #2: Yes

Author’s response: Thank you for your feedback.

2. Is the protocol technically sound and planned in a manner that will lead to a meaningful outcome and allow testing the stated hypotheses?

Reviewer #1: Yes

Reviewer #2: Partly

Author’s response: Thank you for your feedback. We will address this point in detail within the specific feedback (point L) provided by the reviewers below.________________________________________

3. Is the methodology feasible and described in sufficient detail to allow the work to be replicable?

Reviewer #1: Yes

Reviewer #2: Yes

Author’s response: Thank you for your feedback.

4. Have the authors described where all data underlying the findings will be made available when the study is complete?

Reviewer #1: No

Reviewer #2: Yes

Author’s response: Thank you for your feedback. We will address this point in detail within the specific feedback (point K) provided by the reviewers below.

5. Is the manuscript presented in an intelligible fashion and written in standard English?

Reviewer #1: Yes

Reviewer #2: Yes

Author’s response: Thank you for your feedback.

6. Review Comments to the Author

You may also provide optional suggestions and comments to authors that they might find helpful in planning their study.

Author’s response: Thank you for your feedback.

Reviewer #1: The scoping review protocol is an interesting beginning for a research study. I have many minor modifications and few questions about the manuscript. The overall work look interesting and it could provide a good paper to fill the clinical evidence summarizing the information on chrononutrition factors in association with glycemic outcomes in adult population.

a) The introduction (both in abstract and the manuscript) should be enlarged in order to provide larger detail and context of the topic. Actually, is pretty limited.

Why you haven't included grey literature in the studies selected? Please justify it.

Author’s response: Thank you for your feedback. We appreciated your suggestion to expand the introduction. We have revised both abstract and manuscript to provided more detailed context and background on the topic the grey literature in the manuscript.

Abstract, Page 2, Lines 28 to 31: Chrononutrition, which examines the relationship between circadian rhythms and nutrition, has been associated with glycemic outcomes in adults. However, published data on delayed meal timing, increased meal frequency and frequent breakfast skipping have shown inconsistent glycemic outcomes due to variations in methodologies and populations studied.

Introduction, Page 3, Line 54 to Page 5, Lines 119: Diabetes is a lifelong metabolic disease with a rising global prevalence, projected to affect 260.2 million adults in the Western Pacific region by 2045 [1,2]. Despite the American Diabetes Association’s lifestyle recommendations have suggested healthy diet and physical activity to regulate blood glucose levels, but current guidelines may be insufficient for effective diabetes management [3]. Emerging evidence increasingly highlights the critical role of circadian clock in metabolic regulation, introducing chrononutrition as a significant factor in diabetes management [4–6]. Recently, modern adults often engage in irregular eating patterns, including skipping breakfast [7,8], extending daily eating windows [9], reducing nocturnal fasting periods [10], and frequent nighttime snacking [11]. These behaviors are closely related to insulin resistance, leading to elevated glycated hemoglobin levels (HbA1c), increased postprandial and fasting plasma glucose levels, and a higher risk of diabetes [12–16].

Chrononutrition’s key components, including energy distribution, meal regularity and frequency of meals, have been shown to influence glycemic outcomes [17]. Current evidence had extended its components to meal timing, specifically night eating [18]. Large-scale studies have demonstrated the association between eating later in the day and poor glycemic control [15,19]. For example, the National Health and Nutrition Examination Survey (NHANES) in the United State (US) found a significant association between the last meal time before bed and elevated HbA1c levels[15]. However, findings from the Korea National Health and Nutrition Examination (KNHANES) did not show a significant association between evening eating, defined as consuming more than 40% of daily energy between 6.00pm to 9.00pm, and adverse glycemic outcomes [19]. In contrast, a US study suggested that restricting eating to window from 4.00 am to 4.00 pm, with light dinner and energy distribution focused on breakfast and lunch could stabilize glucose levels and reduce fluctuations [20]. This statement is also in agreement with Asian studies, which indicated that hyperglycemia is associated with eating meals after 8.00pm [21,22]. Studies also indicated that fasting gaps allocated before sleep by refraining from late-night dinner eating may regulate blood glucose effectively [23,24]. However, identifying specific time-related dietary factors that contribute to glucose intolerance remains complex, influenced by seasonal variations, daylight exposure and individual lifestyle factors [25–27].

Given meal timing is related to meal frequency, night fasting duration, and energy distribution of a day, this study explores their associations with glycemic outcomes. Previous studies have reported that irregular meals, specifically eating more than four meals or fewer than two meals daily, are associated with an increased risk of diabetes [28–31]. However, other studies have found no significant association between meal frequency and diabetes risk [10,32]. These discrepancies are likely due to variations in meal timing and frequencies, leading to inconsistent glycemic outcomes [29,33]. A systematic review suggests that restricting eating to 2 to 3 meals daily, within a feeding window of less than 10 hours and consuming the last meal 3 to 4 hours before sleep, may effectively manage glucose levels in individuals with type 2 diabetes [34]. However, this strategy’s applicability to other populations, including healthy individuals [35], those who are overweight or obese [36], where the effects of meal timing remain uncertain. Furthermore, breakfast skippers delayed their first meal and reserved majority of their energy intake at evening time has provided mixed results on glycemic outcomes [37,38]. Thus, the available evidence showed inconsistency in the association between chrononutrition factors and glycemic outcomes.

Conventional methods for diagnosing glycemic outcomes, such as random blood glucose, fasting plasma glucose, the 75g oral glucose tolerance test and HbA1c levels may not reliably capture advanced interday or intraday glucose fluctuations [39]. Therefore, this study incorporates studies that using the ambulatory glucose profiles derived from continuous glucose monitoring sensors could provide a more advanced and comprehensive understanding of 24-hour glucose variability in individuals with hyperglycemia [40]. Understanding how chrononutrition factors interact and influence glycemic outcomes is essential, although health effects of combining different chrononutrition factors remain uncertain. A comprehensive evaluation of these factors could provide valuable insights into their association with glycemic outcomes. With the ultimate aim to improve the diabetes management, the present review will identify and analyzes the knowledge gaps regarding the association between chrononutrition factors and glycemic outcomes in adult population. Specifically, this review aims to: 

1. investigate the association between chrononutrition factors (meal regularity, meal timing and meal frequency) and glycemic outcomes in terms of glucose indices, insulin indices and incidence of diabetes among adults.

2. describe the chrononutrition factors (meal regularity, meal timing and meal frequency) applied in various populations and their effect on glycemic outcomes (glucose indices, insulin indices and incidence of diabetes) of the populations.

3. examine the opportunities to integrate chrononutrition into medical nutrition therapy to prevent and control diabetes mellitus. 

b) Why only studies from 2012? Please justify it.

Author’s response: Thank you for your feedback. We intended to select studies from past ten years to capture the most recent and relevant research, reflecting the developments in this rapidly evolving field up to the time of our manuscript preparation and submission in 2024. To ensure our review includes the most up-to-date evidence, we have now revised and extended our search to include studies published from 2014 up to 2024.

Abstract, Page 2, Lines 36 to 38: This review focuses on original articles published from January 2014 to 2024, involving participants aged 18 years and older, published in English, and encompassing experimental and observational studies.

Materials and methods, Step 2, Page 8, Lines 184 to 187: Studies that focus on: (1) adult population aged 18 to 60 years; (2) original journal articles published from 2014 to 2024; (3) the full text available in English will be included and (4) the experimental and observational studies.

c) In the introduction you report that in 2045 there will be 783 millions of people with diabetes. I could suggest also to introduce the actual incidence and prevalence worldwide or in some specific countries (such as US, Europe).

Author’s response: Thank you for your feedback. We have included the projected incidence and prevalence of diabetes for the Western Pacific region in 2045 in the content.

Introduction, Page 3, Lines 54 to 55: Diabetes is a lifelong metabolic disease with a rising global prevalence, projected to affect 260.2 million adults in the Western Pacific region by 2045.

d) When you state that "In recent years, a growing body of evidence suggests that the circadian clock interconnects with nutrients, and impacts on body's physiological process, namely chrononutrition [7]", I expect at least several manuscripts and some systematic literature reviews that state this, not just a study.

Author’s response: Thank you for your feedback. We have included several literature reviews for this statement. 

Introduction, Page 3, Lines 58 to 60: Emerging evidence increasingly highlights the critical role of circadian clock in metabolic regulation, introducing chrononutrition as a significant factor in diabetes management [4–6].

e) In the objective of the scoping review you state that you will provide a definition of chrononutrition characteristics. What will you use for it? Maybe a concept analysis? Please provide larger detail.

Author’s response: Thank you for your feedback. After further discussion with the team, we have decided to exclude the objective of defining chrononutrition characteristics. Instead, we will focus on our primary aim of investigating the association between chrononutrition factors and glycemic outcomes.

Introduction, Page 5, Line 111 to Page 6, Line 119: Specifically, the review aims to: 

1. investigate the association between chrononutrition factors and glycemic outcomes among adults.

2. describe the chrononutrition factors applied in various populations and their effect on glycemic outcomes of the populations.

3. examine the opportunities to integrate chrononutrition into medical nutrition therapy to prevent and control diabetes mellitus. 

f) Can you provide in the supplementary material the data extraction form that you will use in the scoping review?

Author’s response: Thank you for your feedback. We have included the data extraction form in the Supplementary Material 2. 

g) Can you provide larger details of the benefits of this review for government and policymakers?

Author’s response: Thank you for your feedback. We have provided the details of the benefits of this review for government and policymakers. 

Discussion, Page 17, Line 320 to Page 18, Line 334: At population level, the findings of the review are crucial for governments and policymakers. Incorporating circadian eating into clinical practice guideline, dietary recommendations and public health policies aimed at managing glycemic control and preventing diabetes could have significant benefits. Specifically, these guidelines might advocate for consistent daily eating start and end times, ensuring the last meal occurs before a certain clock time or within a defined period before bedtime. By identifying the most impactful chrononutrition factors influencing glycemic outcomes, resources can be efficiently directed towards targeted programs and initiatives, thus maximizing the return on investment in public health. Furthermore, this review has also highlighted the gaps in current research, promoting collaboration between government bodies, academic institutions, and healthcare providers. This cooperation is essential for driving innovation in chrononutrition and metabolic health research. The insights gained can also facilitate international collaborations

---

## [Decision Letter · Decision Letter 1]

23 Sep 2024

PONE-D-23-29067R1Scoping review protocol: The chrononutrition factors in association with glycemic outcomes in adult population.PLOS ONE

Dear Dr. Koo,

Thank you for submitting your manuscript to PLOS ONE. After careful consideration, we feel that it has merit but does not fully meet PLOS ONE’s publication criteria as it currently stands. Therefore, we invite you to submit a revised version of the manuscript that addresses the points raised during the review process.

One Reviewer still had some minor comments. Please address them.;

We look forward to receiving your revised manuscript.

Kind regards,

Yee Gary Ang, MBBS MPH

Academic Editor

PLOS ONE

Journal Requirements:

Additional Editor Comments:

We would like to invite you to make changes before accepting the publication

Reviewers' comments:

Reviewer's Responses to Questions

Comments to the Author

1. Does the manuscript provide a valid rationale for the proposed study, with clearly identified and justified research questions?

Reviewer #1: Yes

Reviewer #2: Yes

2. Is the protocol technically sound and planned in a manner that will lead to a meaningful outcome and allow testing the stated hypotheses?

Reviewer #1: Yes

Reviewer #2: Yes

3. Is the methodology feasible and described in sufficient detail to allow the work to be replicable?

Reviewer #1: Yes

Reviewer #2: Yes

4. Have the authors described where all data underlying the findings will be made available when the study is complete?

Reviewer #1: Yes

Reviewer #2: Yes

5. Is the manuscript presented in an intelligible fashion and written in standard English?

Reviewer #1: Yes

Reviewer #2: Yes

6. Review Comments to the Author

You may also provide optional suggestions and comments to authors that they might find helpful in planning their study.

Reviewer #1: I have only a single question... there is a dedicated PRISMA for protocols of Scoping review. Did you consider this one? In case it is not the one of the Scoping review, please update your supplementary material. On the other hand I am satisfied with the answers provided in the previous review process.

Reviewer #2: The resubmitted manuscript titled "Scoping review protocol: The chrononutrition factors in association with glycemic outcomes in adult population" is well updated. The authors fulfill each comments raised appropriately

7. PLOS authors have the option to publish the peer review history of their article (what does this mean?). If published, this will include your full peer review and any attached files.

Do you want your identity to be public for this peer review? For information about this choice, including consent withdrawal, please see our Privacy Policy.

Reviewer #1: No

Reviewer #2: No

---

## [Author Response · Author response to Decision Letter 1]

23 Sep 2024

Dear reviewers from PLOS One, 

We sincerely thank the reviewers for taking the time to review our manuscript and providing constructive feedback to improve our manuscript. We sincerely hope that you may reconsider our manuscript entitled “Scoping review protocol: The chrononutrition factors in association with glycemic outcomes in the adult population.” (Ref No: PONE-D-23-29076R1) for publication in PLOS One. Enclosed is the revised version of our manuscript. 

Below are our responses to the comments and recommendations.

Reviewers' comments:

Reviewer's Responses to Questions

Comments to the Author

1. Does the manuscript provide a valid rationale for the proposed study, with clearly identified and justified research questions?

Reviewer #1: Yes

Reviewer #2: Yes

Author’s response: Thank you for your feedback.

2. Is the protocol technically sound and planned in a manner that will lead to a meaningful outcome and allow testing the stated hypotheses?

Reviewer #1: Yes

Reviewer #2: Yes

Author’s response: Thank you for your feedback.

3. Is the methodology feasible and described in sufficient detail to allow the work to be replicable?

Reviewer #1: Yes

Reviewer #2: Yes

Author’s response: Thank you for your feedback.

4. Have the authors described where all data underlying the findings will be made available when the study is complete?

Reviewer #1: Yes

Reviewer #2: Yes

Author’s response: Thank you for your feedback.

5. Is the manuscript presented in an intelligible fashion and written in standard English?

Reviewer #1: Yes

Reviewer #2: Yes

Author’s response: Thank you for your feedback.

6. Review Comments to the Author

You may also provide optional suggestions and comments to authors that they might find helpful in planning their study.

Reviewer #1: I have only a single question... there is a dedicated PRISMA for protocols of Scoping review. Did you consider this one? In case it is not the one of the Scoping review, please update your supplementary material. On the other hand I am satisfied with the answers provided in the previous review process.

Author’s response: Thank you for your feedback. We have updated our supplementary material to include the PRISMA-ScR (Preferred Reporting Items for Systematic reviews and Meta-Analyses extension for Scoping Reviews) checklist, which is specifically designed for scoping review protocols. The revised supplementary file has been attached for your review.

Reviewer #2: The resubmitted manuscript titled "Scoping review protocol: The chrononutrition factors in association with glycemic outcomes in adult population" is well updated. The authors fulfill each comments raised appropriately.

Author’s response: Thank you for your feedback.

7. PLOS authors have the option to publish the peer review history of their article (what does this mean?). If published, this will include your full peer review and any attached files.

Do you want your identity to be public for this peer review? For information about this choice, including consent withdrawal, please see our Privacy Policy.

Reviewer #1: No

Reviewer #2: No

Author’s response: Thank you for your feedback.

---

## [Decision Letter · Decision Letter 2]

8 Oct 2024

PONE-D-23-29067R2Scoping review protocol: The chrononutrition factors in association with glycemic outcomes in adult population.PLOS ONE

Dear Dr. Koo,

Thank you for submitting your manuscript to PLOS ONE. After careful consideration, we feel that it has merit but does not fully meet PLOS ONE’s publication criteria as it currently stands. Therefore, we invite you to submit a revised version of the manuscript that addresses the points raised during the review process.

**Please mame the suggested changes before resubmiting.**

We look forward to receiving your revised manuscript.

Kind regards,

Yee Gary Ang, MBBS MPH

Academic Editor

PLOS ONE

**Journal Requirements:**

Reviewers' comments:

Reviewer's Responses to Questions

Comments to the Author

1. Does the manuscript provide a valid rationale for the proposed study, with clearly identified and justified research questions?

Reviewer #1: Yes

Reviewer #2: Yes

2. Is the protocol technically sound and planned in a manner that will lead to a meaningful outcome and allow testing the stated hypotheses?

Reviewer #1: Yes

Reviewer #2: Yes

3. Is the methodology feasible and described in sufficient detail to allow the work to be replicable?

Reviewer #1: Yes

Reviewer #2: Yes

4. Have the authors described where all data underlying the findings will be made available when the study is complete?

Reviewer #1: Yes

Reviewer #2: Yes

5. Is the manuscript presented in an intelligible fashion and written in standard English?

Reviewer #1: Yes

Reviewer #2: Yes

6. Review Comments to the Author

You may also provide optional suggestions and comments to authors that they might find helpful in planning their study.

**Reviewer #1: **I have very few questions:

I think that your review should address all the gaps in the literature and trying to harmonize3 the outcomes and trying to investigate in deep in possible issues derived by discording results. You should provide a larger set of review objectives.

Why do you consider people within 18-60 years ? Please justify it.

**Reviewer #2: **The manuscript title "Scoping review protocol: The chrononutrition factors in association with glycemic outcomes in adult population"" is well revised. The authors fulfill each response appropriately.

7. PLOS authors have the option to publish the peer review history of their article (what does this mean?). If published, this will include your full peer review and any attached files.

Do you want your identity to be public for this peer review? For information about this choice, including consent withdrawal, please see our Privacy Policy.

Reviewer #1: No

Reviewer #2: No

---

## [Author Response · Author response to Decision Letter 2]

15 Oct 2024

Dear reviewers from PLOS One, 

We sincerely thank the reviewers for taking the time to review our manuscript and providing constructive feedback to improve our manuscript. We sincerely hope that you may reconsider our manuscript entitled “Scoping review protocol: The chrononutrition factors in association with glycemic outcomes in the adult population.” (Ref No: PONE-D-23-29076R2) for publication in PLOS One. Enclosed is the revised version of our manuscript. 

Below are our responses to the comments and recommendations.

Reviewers' comments:

1. Does the manuscript provide a valid rationale for the proposed study, with clearly identified and justified research questions?

Reviewer #1: Yes

Reviewer #2: Yes

Author’s response: Thank you for your feedback.

2. Is the protocol technically sound and planned in a manner that will lead to a meaningful outcome and allow testing the stated hypotheses?

Reviewer #1: Yes

Reviewer #2: Yes

Author’s response: Thank you for your feedback.

3. Is the methodology feasible and described in sufficient detail to allow the work to be replicable?

Reviewer #1: Yes

Reviewer #2: Yes

Author’s response: Thank you for your feedback.

4. Have the authors described where all data underlying the findings will be made available when the study is complete?

Reviewer #1: Yes

Reviewer #2: Yes

Author’s response: Thank you for your feedback.

5. Is the manuscript presented in an intelligible fashion and written in standard English?

Reviewer #1: Yes

Reviewer #2: Yes

Author’s response: Thank you for your feedback.

6. Review Comments to the Author

You may also provide optional suggestions and comments to authors that they might find helpful in planning their study.

Reviewer #1: I have very few questions:

I think that your review should address all the gaps in the literature and trying to harmonize3 the outcomes and trying to investigate in deep in possible issues derived by discording results. You should provide a larger set of review objectives.

Why do you consider people within 18-60 years? Please justify it.

Author’s response: Thank you for your valuable feedback. We consider people within 18 to 60 years as adult age range because according to World Health Organization, individuals aged 18 years and older are considered adults in most contexts (WHO, 2015). The age range is important for ensuring that participants are legally capable of providing informed consent and are generally in a period of relatively stable physical and physiological functioning (WHO, 2015). Studies on adults beyond the age of 60 are most likely to be influenced by natural aging processes, such as changes in metabolism, body composition and increased prevalence of chronic diseases can create significant variations in health outcomes (Alan, 2023;Li et al. 2018). Additionally, the working age group (18-60 years) typically represents adults who are actively engaged in employment and have consistent lifestyle patterns related to diet, physical activity and sleep (WHO, 2015; Rudnicka et al. 2020). In line with your suggestion, we also recognize the importance of addressing gaps in the literature by harmonizing findings from discordant results. Our choice of this age range allows us to focus on a more homogenous population, minimizing confounding factors, and thereby improving the precision of our findings. Furthermore, the exclusion of participants over 60 ensure our review objectives remain focused on identifying patterns that are less likely to be influenced by aging-related variations, while also contributing to the larger body of literature that explores health outcomes in this population. 

Citation: 

a) Li Y, Zhao L, Yu D, Wang Z, Ding G (2018) Metabolic syndrome prevalence and its risk factors among adults in China: A nationally representative cross-sectional study. PLOS ONE 13(6): e0199293. https://doi.org/10.1371/journal.pone.0199293.

b) Gutterman, Alan, World Health Organization and Older Persons (January 1, 2023). Available at SSRN: https://ssrn.com/abstract=4320044 or http://dx.doi.org/10.2139/ssrn.4320044

c) World Health Organization. (‎2015)‎. World report on ageing and health. World Health Organization. https://iris.who.int/handle/10665/186463

d) Rudnicka E, Napierała P, Podfigurna A, Męczekalski B, Smolarczyk R, Grymowicz M. The World Health Organization (WHO) approach to healthy ageing. Maturitas. 2020 Sep;139:6-11. doi: 10.1016/j.maturitas.2020.05.018. Epub 2020 May 26. PMID: 32747042; PMCID: PMC7250103.

Pages 6 to 7, lines 141 to 145: We considered individuals aged 18 to 60 years in this group represents a relatively stable period of physical and physiological functioning, minimizing the influence of aging-related factors that could complicate data interpretation. This age group also aligns with the working-age population, which typically has consistent lifestyle patterns related to diet, physical activity and sleep.

Reviewer #2: The manuscript title "Scoping review protocol: The chrononutrition factors in association with glycemic outcomes in adult population"" is well revised. The authors fulfill each response appropriately.

Author’s response: Thank you for your feedback.

7. PLOS authors have the option to publish the peer review history of their article (what does this mean?). If published, this will include your full peer review and any attached files.

Do you want your identity to be public for this peer review? For information about this choice, including consent withdrawal, please see our Privacy Policy.

Reviewer #1: No

Reviewer #2: No

Author’s response: Thank you for your feedback.

---

## [Decision Letter · Decision Letter 3]

4 Nov 2024

Scoping review protocol: The chrononutrition factors in association with glycemic outcomes in adult population.

PONE-D-23-29067R3

Dear Dr. Koo,

We’re pleased to inform you that your manuscript has been judged scientifically suitable for publication and will be formally accepted for publication once it meets all outstanding technical requirements.

Kind regards,

Yee Gary Ang, MBBS MPH

Academic Editor

PLOS ONE

Additional Editor Comments (optional):

Reviewers' comments:

Reviewer's Responses to Questions

**Comments to the Author**

1. Does the manuscript provide a valid rationale for the proposed study, with clearly identified and justified research questions?

Reviewer #1: Yes

2. Is the protocol technically sound and planned in a manner that will lead to a meaningful outcome and allow testing the stated hypotheses?

Reviewer #1: Yes

3. Is the methodology feasible and described in sufficient detail to allow the work to be replicable?

Reviewer #1: Yes

4. Have the authors described where all data underlying the findings will be made available when the study is complete?

Reviewer #1: Yes

5. Is the manuscript presented in an intelligible fashion and written in standard English?

Reviewer #1: Yes

6. Review Comments to the Author

You may also provide optional suggestions and comments to authors that they might find helpful in planning their study.

Reviewer #1: I am satifsfied with the modifications provided. The reviewers provided many requested that improved the quality of the manuscript from its first version

7. PLOS authors have the option to publish the peer review history of their article (what does this mean?). If published, this will include your full peer review and any attached files.

Reviewer #1: No

---

## [Editor Report · Acceptance letter]

26 Nov 2024

PONE-D-23-29067R3 

PLOS ONE

Dear Dr. Koo, 

I'm pleased to inform you that your manuscript has been deemed suitable for publication in PLOS ONE. Congratulations! Your manuscript is now being handed over to our production team.

Kind regards, 

on behalf of

Dr. Yee Gary Ang 

Academic Editor

PLOS ONE